# Posterior Reversible Encephalopathy Syndrome in a Pediatric Patient with End-Stage Renal Disease

**DOI:** 10.3390/children10040731

**Published:** 2023-04-15

**Authors:** Ionela-Loredana Popa, Mihaela Bălgrădean, Mariana Costin, Anca Bobircă, Cristina Bologa, Teodora Armășelu, Florin Bobircă, Anca Croitoru

**Affiliations:** 1Department of Pediatric Nephrology, “M.S. Curie” Emergency Clinical Hospital for Children, 077120 Bucharest, Romania; loredana.popa@drd.umfcd.ro (I.-L.P.);; 2Department of Pediatrics, “Carol Davila” University of Medicine and Pharmacy, 050474 Bucharest, Romania; 3Department of Internal Medicine and Rheumatology, “Carol Davila” University of Medicine and Pharmacy, 050474 Bucharest, Romania; 4Internal Medicine and Rheumatology Department, Dr. Ion Cantacuzino Clinical Hospital, 011437 Bucharest, Romania; 5Department of Surgery, “Carol Davila” University of Medicine and Pharmacy, 050474 Bucharest, Romania; 6Surgery Department, Dr. Ion Cantacuzino Clinical Hospital, 011437 Bucharest, Romania

**Keywords:** end-stage renal disease, pediatric, hypertension, posterior reversible encephalopathy

## Abstract

Posterior reversible encephalopathy syndrome (PRES) is a clinical and neuroimaging syndrome that can affect both children and adults and has variable etiology. It is clinically defined by headaches, consciousness disorders, seizures and visual disturbances. Early recognition (clinical and imaging) can lead to appropriate general measures to correct the underlying cause of PRES. In this paper, we report a case of PRES in an eight-year-old boy with bilateral renal hypoplasia and end-stage renal disease (ESRD).

## 1. Introduction 

Posterior reversible encephalopathy syndrome (PRES), also known as reversible posterior leukoencephalopathy syndrome (RPLS), is observed in different age groups, including pediatric patients. Usually, if the syndrome is recognized and promptly treated, the patients recover clinically in days or weeks with no neurological side effects [1,2].

PRES occurs when posterior circulation is unable to autoregulate due to acute changes in blood pressure. This neurotoxic state, secondary to hyperperfusion and the disruption of the blood–brain barrier, results in vasogenic edema, mostly described in parieto-occipital regions (70–90% of cases). PRES can be found in non-posterior regions, mainly in watershed areas such as the frontal, inferior-temporal, cerebellar and brainstem regions [3]. In 5% of cases, the pattern of PRES is uncommon and can be unilateral in the central brainstem or basal ganglia or may not include cortical or subcortical white matter involvement in the spinal cord. The literature describes a female predominance in the patient population, implying that some underlying comorbidities are gender-specific [4,5]. PRES can occur secondary to different conditions, including hypertension, systemic infections, COVID-19 [6,7], autoimmune diseases (systemic lupus erythematosus or Wegener’s granulomatosis), hemolytic uremic syndrome, thrombocytopenic thrombotic purpura, sickle cell disease [8], ventriculoperitoneal shunt insertion/overshunting [9], malignant tumors, chemotherapy or immunosuppression and drug toxicity (calcineurin inhibitors, cyclophosphamide, tacrolimus, azathioprine, erythropoietin, L-asparaginase and filgrastim) [10,11,12]. The most common clinical signs are headaches, seizures, acute confusion or an altered mental state and visual disturbances such as reversible cortical blindness [3,13]. Other neurological symptoms may include focal neurological deficits, ataxia and vertigo [14]. Depending on the radiologic features, different examinations can be performed. On computed tomography, the affected regions are hypoattenuated, and on angiography, signs of vasospasm or arteritis are observed, i.e., diffuse or focal vasoconstriction, vasodilation and a string-of-beads appearance. Brain magnetic resonance imaging (MRI) shows T1 hypointense and T2 hyperintense lesions in the affected regions. Pediatric patients with acute or chronic kidney disease are at risk of neurological complications secondary to hemodynamic or electrolyte instability and chronic therapies (dialysis and medication). There are a few articles which present recurrent PRES episodes, especially in patients undergoing dialysis [15,16,17]. For children with PRES, there is no specific treatment; the underlying condition is managed with supportive treatment [18]. In this article, we present the case of a patient with ESRD who developed PRES secondary to hypertension.

## 2. Case Report

An eight-year-old Caucasian boy with no personal pathological antecedents (data are inconclusive) was diagnosed with bilateral renal hypoplasia and ESRD in our Department of Pediatric Nephrology in April 2022. On admission, he was transferred to our unit for marked pallor; weight gain; general edema; elevated serum creatinine (8.14 g/dL), serum urea (203 mg/dL) and uric acid (7.6 mg/dL) levels; severe normocytic and normochromic anemia (Hb—4.3 g/L); and a high blood pressure (BP) of >138/80 mmHg.

Investigations highlighted the level of kidney function and damage, thus assessing the effects on the rest of the body systems.

From the assessment of kidney function (Table 1), the patient was diagnosed with stage 5 end-stage renal disease with an estimated glomerular filtration rate (eGFR) of 5 mL/min/1.73 m^2^ (creatinine-based “Bedside Schwartz” equation). An abdominal ultrasound revealed significant kidney damage: bilateral renal hypoplasia (right kidney—5.1/3.6/2.9 cm, left kidney—5.3/3.1/2.9 cm) with low cortico-medullary differentiation.

Knowing the effects of ESRD, further evaluation examined the patient’s acid–base balance and blood, cardiac, skeletal and endocrine systems. The acid–base balance showed metabolic acidosis (pH of 7.31) with bicarbonate deficiency ((HCO3) of 13.7 mEq/L) that was unresponsive to alkaline therapy, but normal sodium (139 mg/dL) and potassium (3.5 mg/dL) levels.

A damaged kidney affects the production of the hormone erythropoietin, a process that explains the patient’s severe normocytic and normochromic anemia (hemoglobin = 4.3 g/dL) with low transferrin saturation (<10%), high serum ferritin (326.5 ng/mL) and normal serum iron (75.62 mcg/dL). To correct the anemia, the patient received a blood transfusion (hemoglobin = 6 g/dL) and continued with intravenous erythropoietin.

Continuing investigation into the other body systems, echocardiography showed second-grade aortic insufficiency, left ventricular hypertrophy (LVH) and pericarditis (Table 2). The high N-terminal pro-B-type natriuretic peptide (NT-proBNP) in ESRD is due to decreased renal elimination, fluid overload, hypertension and LVH, all of which the patient presented.

The series of changes that occur in chronic kidney diseases (CKD) such as ESRD, including a decrease in calcium levels due to impairment in vitamin D activation, high phosphate levels and the hypersecretion of parathyroid hormone (PTH, secondary hyperparathyroidism), are defined as CKD mineral bone disease and are presented in Table 3.

Furthermore, the patient presented with primary hypothyroidism; laboratory markers evidenced significant levels of thyroid hormone levels (TSH = 2243 mIU/L) and low triiodothyronine (T3), thyroxine (T4), freeT3 and freeT4 (Table 4). He was started on levothyroxine with increasing doses.

After assessing the patient’s diagnosis (ESRD with secondary hypertension, LVH, primary hypothyroidism, secondary hyperparathyroidism and chronic anemia), chronic renal replacement therapy (hemodialysis) was initiated. The appropriate chronic hemodialysis treatment was three sessions/week, for four hours each session (ultrafiltration rate 150 mL/h) with an intravenous erythropoietin-stimulating agent (darbepoetin alfa). It is well known that stage 5 chronic kidney disease is associated not only with hypertension but also with cardiovascular disease (adverse cardiovascular events). The underlying mechanisms cited in patients undergoing hemodialysis are volume overload and sodium retention, arterial stiffness, activation of the renin–angiotensin–aldosterone system, the activation of the sympathetic nervous system and secondary hyperparathyroidism. Taking into consideration all the underlying mechanisms of hypertension during his first admission, our patient was prescribed and sent home with the following antihypertensive treatment: a calcium channel blocker (amlodipine 4 mg/kg/day), a cardioselective beta-1-adrenergic receptor inhibitor (metoprolol 0.5 mg/Kg/day), an ACE inhibitor (enalapril 0.1 mg/kg/day) and a loop diuretic (furosemide 3 mg/kg/day).

Three months later, the patient was hospitalized for persistent headaches, tonic–clonic left-sided seizures and clouding of consciousness, with periods of hypoactivity and agitation and a high blood pressure (>180/96 mmHg), which were maintained both before and after the hemodialysis session. The interval from the initiation of hypertensive treatment and hemodialysis to the current episode was free of any cardiovascular events, with blood pressure values maintained in the range of 120–135/65–70 mmHg. In dialyzed patients, the goal is usually to maintain blood pressure in the normal range, and in many cases, this is difficult to achieve.

A computer tomography (CT) scan was performed on the patient’s brain to exclude vascular etiology (border zone infarcts, cerebral sinovenous thrombosis and intracerebral hemorrhage), and the results were normal. Additionally, a laboratory investigation did not reveal changes that could explain the neurological disorders—no electrolytes (serum sodium: 138 mg/dL, serum potassium: 3.4 mg/dL and ionic calcium: 4 mg/dL) and normal glucose levels. At the time of his second admission, the patient’s hypertensive treatment was adjusted following a multidisciplinary consultation between the pediatric nephrology team and the pediatric cardiology team.

As a result, in the first phase, the initial treatment was escalated to the maximum allowed dosage, but the patient’s blood pressure values remained close to high values of systolic blood pressure ((SBP) >200) and diastolic blood pressure ((DBP) >120 mmHg). The latest research at the time showed that the recommendations for antihypertensive drugs in CKD dialysis patients were based on their BP reduction effects, side effects and protective cardiovascular effects [19]. The pediatric cardiology team decided to also introduce the following antihypertensive drugs: methyldopa (13 mg/kg) and an α₂-adrenergic agonist (clonidine 150 mcg/kg). Erythropoietin-stimulating agents may also play an important role in the etiology of hypertension in dialysis patients; thus, the treatment was changed to subcutaneous administration. The maximum doses for treating hypertension were prescribed without any response, and the clinicians decided to start the continuous intravenous infusion of nicardipine. After 48 h, because there were no changes in his clinical status and the values of his blood pressure did not drop below SBP > 200 and DBP > 120 mmHg, the continuous intravenous infusion of urapidil 0.8 mg/kg was initiated, and a favorable response was achieved safely without any other side effects.

Before starting treatment with urapidil, the patient’s mental status started to decline; he experienced impaired awareness, confusion and trouble speaking, and the team decided to perform brain magnetic resonance imaging (MRI). The MRI showed islands of hypersignals diffusely outlined and mostly located above the occipital supratentorial region, but also frontally with a subcortical disposition. It showed bilateral damage of the fibers in a “U” shape located at the level of the posterior cerebellar fossa, with larger dimensions on the right-side restriction of the gyri. Linear diffusion was also observed in a few occipital lesions and in right cerebellar partial lesions without degradation products of hemoglobin. The angio-type sequence with fast flow highlighted the gracil flow in the communicator on the right side of the brain (Figure 1, Figure 2 and Figure 3).

No susceptibility artifacts indicating an underlying hemorrhage or underlying mass lesions were identified.

The clinical manifestations and MRI imaging confirmed the suspicion of PRES. Treatment with urapidil via continuous infusion was maintained for 7 days, with an improvement in neurological symptomatology and a decrease in BP values to 120–135/70–90 mmHg. The patient’s state of consciousness was preserved without headaches or other paroxysmal manifestations, and he continued with antihypertensive treatment, aiming to pharmacologically control hypertension.

For the differential diagnosis, we considered systemic vasculitis and autoimmune and infection etiologies. The team decided to perform the following tests for systemic vasculitis (Wegener vasculitis) and autoimmune diseases (systemic lupus erythematosus, Sjogren syndrome, scleroderma, juvenile idiopathic arthritis and autoimmune encephalitis): C3, C4, antineutrophil cytoplasmatic antibodies (ANCA), anti-double-stranded DNA (anti-dsDNA), lupus anticoagulant, antinuclear antibodies (ANA), ANA immunoblot (anti-Ro/SSA antibodies and anti-La SSB), immunogram, antiphospholipid profile, rheumatoid factor, cyclic citrullinated peptide antibodies and *N*-methyl-*D*-aspartate receptor (NMDA). All results were in the normal range.

Viral serology for cytomegalovirus, Epstein–Barr, hepatitis B and C and human immunodeficiency virus were negative and lumbar puncture was inconclusive. COVID-19 is also associated with PRES, but all our patients are tested with rapid and polymerase chain reaction (PCR) tests upon admission to the department, and in this case, the result was negative. It is important to mention that the patient was not undergoing any immunosuppressive or steroid treatments. We proposed to test him for genetic causes of kidney disease, but the parents refused.

Parasitology tests (echinococcus granulosus, taenia solium, Toxocara canis, toxoplasma gondii and trichinella spiralis), conducted due to his socioeconomic environment, were negative.

Electroencephalography (EEG) showed a normal alpha rhythm with no underlying background abnormalities.

Six months after the episode, we repeated the MRI, which showed a complete reversal of the described lesions. The patient recovered completely.

## 3. Discussion

PRES should be considered in all patients with kidney disease, especially those with ESRD who present with hypertension and endocrine abnormalities.

The association between hypertension and cardiovascular disease risk has been well documented in the general CKD population, but in dialysis patients, the associated risk is higher and poorly understood. Hypertension (blood pressure > 130/85 mm Hg) is common in patients undergoing regular dialysis, with a prevalence of 70–80% among regular hemodialysis patients; in addition, only a minority have adequate blood pressure control. Volume overload and sodium retention is considered the main pathogenic mechanism of hypertension in hemodialytic patients, but other factors are also involved, such as arterial stiffness/endothelial disfunction, the activation of the renin–angiotensin–aldosterone system and the sympathetic nervous system, the use of recombinant erythropoietin and sleep apnea [20].

Analyzing our case and other cases of ESRD, PRES and uremic encephalopathy, the main causes taken into consideration are retention of uremic toxins (urea, guanidines, oxalate, trimethylamine-N- oxide, phosphorus, hydrogen ion, P-cresol and p-cresyl sulfate, homocysteine, tryptophan metabolites, phenyl acetic acid, beta2- microglobulin, PTH and advanced glycosylation end products), alterations in hormonal metabolism, changes in electrolyte and acid-base homeostasis, increased vascular reactivity, blood–brain barrier transport, chronic inflammation and imbalances of neurotransmitter amino acids within the brain. The progression of uremia determines the accumulation of guanidino compounds, resulting in activation of excitatory N-methyl-D-aspartate (NMDA) receptors and inhibition of inhibitory glutamine and gama-aminobutyric acid (GABA) receptors, which lead to seizures and myoclonus. Alterations in the metabolism of dopamine and serotonin in the brain can cause sensorial clouding. Manifestations can also occur during or immediately after dialysis, because urea is rapidly removed during hemodialysis. The rapid decline in the blood level of urea lowers plasma osmolality, creating a transient osmotic gradient between plasma and brain cells, which leads to cerebral edema. During dialysis, where urea is swiftly moved out of circulation, its continued presence in tissues, including brain cells, may determine an osmotic force, drawing water into the cells and producing cerebral edema. Additionally, uremic toxins involving endothelial dysfunction are responsible for the development and progression of the disease. Multiple cytokines, such as interleukin-1 and tumor necrosis factor, are elevated in ESRD and contribute to the pathogenesis of PRES.

ESRD, primary hypothyroidism and secondary hyperparathyroidism, all of which the patient presented with, are factors that determine secondary hypertension that is resistant to pharmacologic therapy, increasing the patient’s risk for developing PRES. The mechanisms for determining secondary hypertension are different. Secondary hyperparathyroidism is responsible for secondary hypertension through different mechanisms, such as determining calcium deposits in the vessel walls leading to vascular stiffness; stimulating PTH2 receptors’ expression in vascular smooth muscle cells; increasing collagen deposits; inducing ventricular hypertrophy with a reduction in myocardial contractility; and causing an increase in heart rate. Hypertension, mostly diastolic hypertension, is increased in patients with hypothyroidism because of the increased peripheral vascular resistance by decreasing the release of the endothelial-derived relaxing factor, promoting the contraction of the smooth muscle cells. To be sure of the causes of PRES, we also excluded other conditions associated with it, such as autoimmune diseases, systemic vasculitis and infectious diseases. The mechanism of PRES is not fully established, but hypertension plays a significant role, along with other hypotheses. The first theory is that high blood pressure leads to a loss of self-regulation, hyperperfusion with endothelial damage and vasogenic edema. The other theories are that vasospasm results in local ischemia and hypoperfusion and that endothelial dysfunction is secondary to circulating endogenous or exogenous toxins [19,21,22].

In children, the incidence of PRES is between 0.04% and 5.2% [4,17,23]. In most cases, seizures and headaches are the most common presentation symptoms. In patients with hypertension, blood pressure should be reduced to no more than 20–25% in the first few hours to avoid renal, coronary and cerebral ischemia [24]. In our case, the predominant symptoms were a persistent headache, tonic–clonic left-sided seizures and clouding of consciousness, with periods of hypoactivity and agitation. The first imaging test for our case was a brain CT with no abnormalities, followed by an MRI with fronto-parietal, occipital and cerebellar regions affected. MRI is the most useful investigation as in most cases, the T1-weighted hypointense, T2-weighted hyperintense and T2-weighted FLAIR hyperintense areas are revealed bilaterally in the occipital and parietal lobes. Vasogenic edema is considered responsible for the pathophysiology of PRES, and is, in most cases, reversible [25,26,27,28].

Not only it is important to differentiate PRES from other causes, but also, the most important first step in treating PRES is the prompt removal of underlying causes, which in our case, was severe hypertension. PRES may complicate a range of underlying comorbidities among the pediatric population. PRES is reversible when the precipitating cause is eliminated or treated, thus avoiding permanent neurological damage and reducing morbidity or eventual mortality.

Recurrent PRES is observed in 4% of patients [4,29,30], especially in those with persistent risk factors such as sickle cell crises, autoimmune conditions, hypertensive crises and renal failure [31,32,33].

## 4. Conclusions

Upon analyzing this case and the literature, we conclude that PRES is a syndrome that must be taken into consideration in pediatric patients with seizures, altered consciousness and accompanying systemic hypertension at onset. Patients with chronic renal disease, especially ESRD, are more predisposed to PRES because they are prone to secondary hyperparathyroidism and hypothyroidism, which are factors leading to secondary hypertension.

Mainly, the mechanism of PRES is vasogenic edema, in which hypertension overwhelms the ability of the cerebrovascular auto-regulation system, resulting in disruption of the blood–brain barrier and capillary leakage. The second mechanism involves cytotoxic edema followed by vasospasm and tissue ischemia.

All patients with ESRD present with uremic toxins, alterations in hormonal metabolism, changes in electrolyte and acid-base homeostasis, increased vascular reactivity, blood–brain barrier transport, chronic inflammation and an imbalance of neurotransmitter amino acids within the brain. All these changes eventually lead to PRES due to vasogenic edema or cytotoxic edema, determining vasospasm and tissue ischemia.

The diagnosis of PRES is easy to determine by an MRI brain scan, where specific modifications are described in the frontal and parieto-occipital regions. Its treatment involves correcting the triggers, in our case hypertension, and supporting patients through the acute phase of the illness. The case presented had a good outcome, because the patient did not have any neurological sequalae, despite the affected regions (fronto-parietal, occipital and cerebellar areas), and the MRI six months later was normal in the regions affected. Usually, PRES is a benign condition with a good prognosis.

## Figures and Tables

**Figure 1 children-10-00731-f001:**
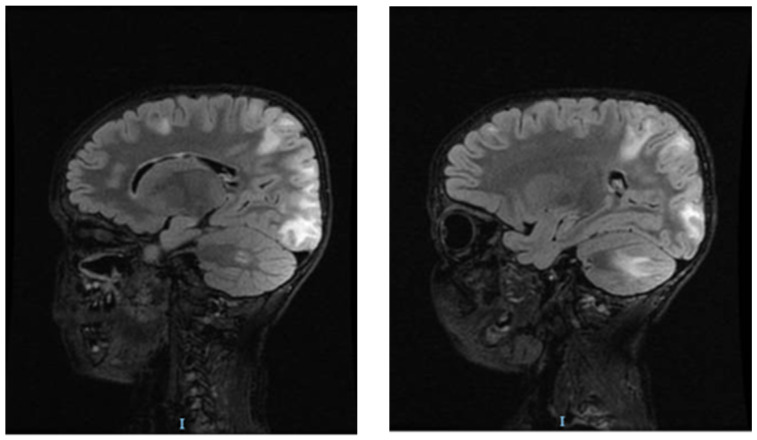
Brain MRI of patient with hyperintensities specific for PRES: T2 FLAIR modifications in the parieto-occipital regions.

**Figure 2 children-10-00731-f002:**
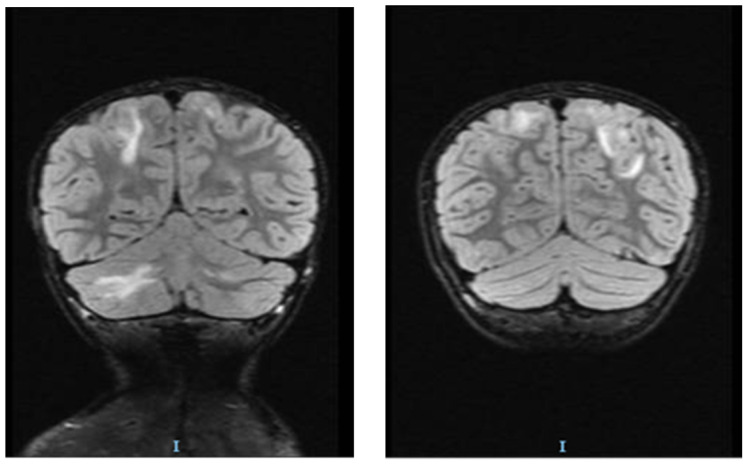
Brain MRI of patient with hyperintensities specific for PRES: coronal FLAIR modifications in the bilateral frontal and left cerebellar regions.

**Figure 3 children-10-00731-f003:**
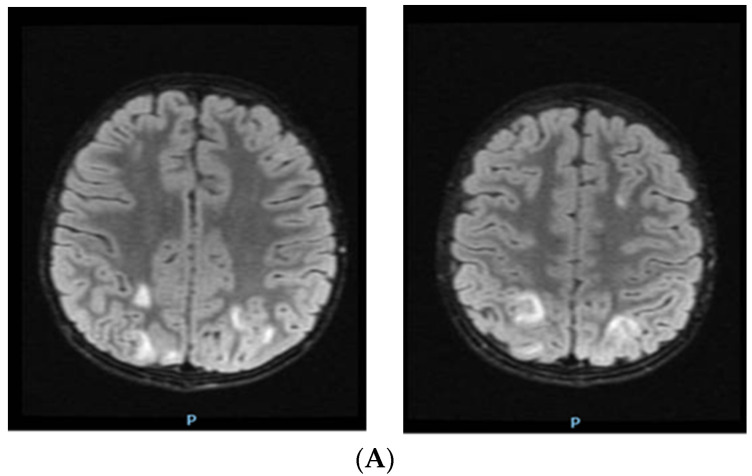
Brain MRI of patient with hyperintensities specific for PRES: (**A**) axial FLAIR modifications in the bilateral occipital region; (**B**) axial FLAIR modifications in the cerebellar region.

**Table 1 children-10-00731-t001:** Kidney laboratory parameters on admission.

Parameters	Results	References
Kidney Function		
Serum creatinine	8.14 mg/dL	0.3–0.7 mg/dL
Serum urea	203 mg/dL	5–20 mg/dL
Uric acid	7.3 mg/dL	<6 mg/dL
BUN	94.74 mg/dL	
BUN—nitrogen content of urea		

**Table 2 children-10-00731-t002:** Cardiac laboratory parameters on admission.

Parameters	Results	References
Cardiac Enzymes		
NT-proBNP	9176 pg/mL	<125 pg/mL
CK-MB	19 U/L	5–25 U/L
CK	531 UI/L	<250 U/L

NT-proBNP: N-terminal pro B type natriuretic peptide.

**Table 3 children-10-00731-t003:** Phospho-calcium balance.

Parameters	Results	References
Ionic calcium	2.0 mg/dL	3.8–5.6 mg/dL
Serum calcium	4.33 mg/dL	8.8–10.8. mg/dL
Serum phosphate	8.41 mg/dL	3.1–5.5 mg/dL
PTH	1172 pg/mL	14–65 pg/mL
Vitamin D	10.6 ng/mL	≥30 ng/mL
PTH—parathyroid hormone.		

**Table 4 children-10-00731-t004:** Thyroid laboratory parameters on admission.

Thyroid Function		
freeT3	1.05 pg/mL	2.53–5.22 pg/mL
freeT4	0.24 ng/mL	0.97–1.67 ng/mL
T3	0.2 pg/mL	0.93–2.31 pg/mL
T4	0.31 ng/mL	5.99–13.8 ng/mL
TSH	2243 mIU/L	0.6–4.84 mIU/L
TPO	11.8 U/mL	0.00–18 U/mL

TSH: thyroid hormone; T3: triiodothyronine; T4: thyroxine; ATPO: antithyroid peroxidase.

## Data Availability

Not applicable.

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
