# Peer review of "Posterior Reversible Encephalopathy Syndrome in a Pediatric Patient with End-Stage Renal Disease"

_children, 2023, doi:10.3390/children10040731_

Round 1

Reviewer 1 Report

Article entitled “Posterior Reversible Encephalopathy Syndrome in a Pediatric 2 Patient with End-Stage Renal Disease” it’s very interesting, written in a clear and easy to read way. In the introduction, among the listed reasons, the COVID-19 should be included, cases of PRES have been described in this group of patients. Moreover, in the description of the case in line 57 -58 there is no exact name of the group of drugs administered to the child. The authors only wrote that they used "specific medications for hypothyroidism, antihypertensive medications".
I suggest writing exactly what drugs were used.

Author Response

Reviewer comment #1: In the introduction, among the listed reasons, the COVID-19 should be included, cases of Pres have been described in this group of patients. Moreover, in the description of the case in line 57-58 there is no exact name of the group of drugs administered to the child. The authors only wrote that they used "specific medications for hypothyroidism, antihypertensive medications"

Response: Thank you for your comment. We added more information in the introduction as suggested, including COVID-19.

Text now reads (Line 98-101): Furthermore, the patient presented with primary hypothyroidism; laboratory markers evidenced significant levels of thyroid hormone levels (TSH= 2243 mIU/L) and low tri-iodothyronine (T3), thyroxine (T4), freeT3 and freeT4 (Table 4). He was started on levothyroxine, with increasing doses.

Text now reads (Line 114-119): Taking into consideration all the underlying mechanisms of hypertension during his first admission, our patient was prescribed and sent home with the following antihypertensive treatment: calcium channel blocker (Amlodipine 4mg/kg/day), cardioselective beta-1-adrenergic receptor inhibitor (Metoprolol 0,5 mg/Kg/day), ACE inhibitor (Enalapril 0,1mg/kg/day) and loop diuretic (Furosemide 3mg/kg/day).

Reviewer 2 Report

Popa et al. reported a child patient of posterior reversible encephalopathy syndrome (PRES) with end-stage renal disease. This is a very typical case and very valuable. However, there are several points that should be addressed:

1.     The authors have to exclude autoimmune diseases such as lupus, Sjogren's syndrome, scleroderma, autoimmune encephalitis, etc., as these diseases may concurrently cause neurological and renal diseases.

2.     Please provide a history of previous medications, such as steroids or immunosuppressants, to exclude drug-related factors.

3.     The authors need to provide the patient's electrolyte (sodium, potassium) levels to determine whether this is the demyelinating lesion caused by electrolyte imbalance.

4.     Please provide further information on whether the patient is undergoing dialysis.

5.     The authors need to provide whether the patient has a genetic background related to primary kidney disease to determine if there is indeed a correlation between the two diseases.

Author Response

Reviewer comment #1: The authors have to exclude autoimmune diseases such as lupus, Sjogren's syndrome, scleroderma, autoimmune encephalitis, etc., as these diseases may concurrently cause neurological and renal diseases

Response: Thank you for your comment. We made the recommended suggestions.

Text now reads (lines 182-190): For the differential diagnosis, we considered systemic vasculitis and autoimmune and infection etiologies. The team decided to perform the following tests for systemic vas-culitis (Wegener vasculitis) and autoimmune disease (systemic lupus erythematosus, Sjogren syndrome, scleroderma, juvenile idiopathic arthritis, autoimmune encephalitis): C3, C4, antineutrophil cytoplasmatic antibodies (ANCA), anti-double-stranded DNA (anti-dsDNA), lupus anticoagulant, antinuclear antibodies (ANA), ANA immunoblot (anti-Ro/SSA antibodies and anti-La SSB), immunogram, antiphospholipid profile, rheumatoid factor, cyclic citrullinated peptide antibodies and N-methyl-D-aspartate receptor (NMDA). All results were in the normal range.

Reviewer comment #2: Please provide a history of previous medications, such as steroids or immunosuppressants, to exclude drug-related factors.

            Response: We thank the reviewer for the insight and added the missing information.

Text now reads (lines 195-196): It is important to mention that the patient was not undergoing any immunosuppressive or steroid treatments

Reviewer comment #3: The authors need to provide the patient's electrolyte (sodium, potassium) levels to determine whether this is the demyelinating lesion caused by electrolyte imbalance.

           Response: We thank the reviewer for the insight and completed the information with the values of the patient electrolyte

Text now reads (lines 130-133): Additionally, a laboratory investigation did not reveal changes that could explain the neurological disorders—no electrolytes (serum sodium: 138 mg/dL; serum potassium: 3.4 mg/dL and ionic calcium: 4 mg/dL) and normal glucose levels.

Reviewer comment #4: Please provide further information on whether the patient is undergoing dialysis.

        Response: We thank the reviewer for the insight and update the information.

Text now reads (lines 107-109): The appropriate chronic hemodialysis treatment was three sessions/week, four hours each session (ultrafiltration rate 150 ml/h) and intravenous erythropoietin-stimulating agent (darbepoetin alfa).

          Reviewer comment #5: The authors need to provide whether the patient has a genetic background related to primary kidney disease to determine if there is indeed a correlation between the two diseases.

         Response: We thank the reviewer for the insight. The parents refused to do any genetic testing.

Text now reads (lines 196-197): We proposed to test him for genetic causes of kidney disease but the parents refused.

Reviewer 3 Report

Overall, the case report lacks supporting evidence to make the connection between PRES and ESRD based on one pediatric patient. Also, the SO WHAT question was not addressed why this information is so important. 

The introduction needs to be improved to flow better and correct grammatical mistakes. Also, several sentences read as though the writer is speaking in the first person and not in a scientific tone. Ex: For children with PRES there is no specific treatment, we manage the underlying condition by offering supportive treatment [12]. We should not be used when writing an introduction. There are several instances of this example throughout the manuscript. 

Table 1 is big and needs to be condensed into a smaller table. Also, the table needs a legend for abbreviations and pertinent information. 

Page three has 5 paragraphs, three of which are too short. Consider combining the information into fewer paragraphs and consolidating your information. Also the same with page 5. 

Author Response

Reviewer comment #1: Table 1 is big and needs to be condensed into a smaller table. Also, the table needs a legend for abbreviations and pertinent information.

Response: Thank you for your comment. We made the recommended suggestions and the table was divided in 3 smaller tables and commented the changes we observed.

Reviewer comment #2: Page three has 5 paragraphs, three of which are too short. Consider combining the information into fewer paragraphs and consolidating your information. Also, the same with page 5.

Response: Thank you for your comment. We added more information to the manuscript according to your suggestions.

Round 2

Reviewer 2 Report

Thanks to the author for adding a lot of data and answering many of my questions. Right now, there are two points need to address:

1. There is a certain correlation between dialysis and uremic encephalopathy. I think the author should discuss the relationship between end-stage renal disease, PRES, and uremic encephalopathy in the Discussion section.

2. Erythropoietin-stimulating agent injections can induce nerve demyelination, I think the authors should discuss erythropoietin and dialysis-induced nerve demyelination in the Discussion section.

Author Response

Reviewer comment #1:  There is a certain correlation between dialysis and uremic encephalopathy. I think the author should discuss the relationship between end-stage renal disease, PRES, and uremic encephalopathy in the Discussion section.

Response: Thank you for your comment. We discussed the relationship between ESRD, PRED and uremic encephalopathy.

Text now reads (lines 275-295 with Track changes or lines 218-238 without Track changes):  Analyzing our case and other cases of ESRD, PRES, and uremic encephalopathy, the main causes taken into consideration are: retention of uremic toxins (urea, guanidines, oxalate, trimethylamine-N- oxide, phosphorus, hydrogen ion, P-cresol and p-cresyl sulfate, homocysteine, tryptophan metabolites, phenyl acetic acid, beta2- microglobulin, PTH, advanced glycosylation end products), alterations in hormonal metabolism, changes in electrolyte and acid-base homeostasis, increased vascular reactivity, blood brain barriers transport, chronic inflammation and imbalances of neurotransmitter amino acids within the brain. The progression of uremia determines the accumulation of guanidino compounds, resulting in activation of excitatory N-methyl-D-aspartate (NMDA) receptors and inhibition of inhibitory glutamine and gama-aminobutyric acid (GABA) receptors, which leads to seizures and myoclonus. Alterations in the metabolism of dopamine and serotonin in the brain can cause sensorial clouding. Manifestations can also occur during or immediately after dialysis because urea is rapidly re-moved during hemodialysis. The rapid decline in the blood level of urea lowers plasma osmolality, creating a transient osmotic gradient between plasma and brain cells, which leads to cerebral edema. During dialysis, where urea is swiftly moved out of the circu-lation, its continued presence in tissues, including brain cells, may determine an osmotic force, drawing water into the cells and producing cerebral edema. Also, uremic toxins involving endothelial dysfunction are responsible for the development and progression of the disease. Multiple cytokines, such as interleukin-1 and tumor necrosis factor, are elevated in ESRD and contribute to the pathogenesis of PRES.

Reviewer comment # 2:  Erythropoietin-stimulating agent injections can induce nerve demyelination; I think the authors should discuss erythropoietin and dialysis-induced nerve demyelination in the Discussion section.

Response: Thank you for your comment. From our knowledge erythropoietin-stimulating agents help in the uremic neuropathy and reduce nerve demyelination. Is that what the reviewer referred to?

Reviewer 3 Report

The case study has been improved. However, the results section and discussion sections need to be improved further. 

Lines 229-261 should not be presented this way. Put the information into a table or write it correctly in sentence format. 

The discussion needs to be presented in more complete paragraphs. There are eight paragraphs on one page! I mentioned that this needed to be changed in my previous comments and edits. 

Author Response

Reviewer comment # 1:  Lines 229-261 should not be presented this way. Put the information into a table or write it correctly in sentence format. 

Response: Thank you for your comment. We consider referring to the differential diagnosis and recalling some of the analyses necessary for confirmation as a reminder to take in consideration. Why do you consider it is not correctly written in sentence format?

Reviewer comment # 2:  The discussion needs to be presented in more complete paragraphs. There are eight paragraphs on one page! I mentioned that this needed to be changed in my previous comments and edits. 

Response: We thank the reviewer for the insight. We tried to have more complete paragraphs. If this is not right, please detail what aspects should we work on more.

    In addition to the above comments, all spelling and grammatical errors have been corrected. We are grateful once again for your consideration of this manuscript. We would like to thank the reviewers for their comments which allowed us to substantially improve the quality of our manuscript. We hope you will find the revised manuscript plausible and will consider it for publication in your esteemed journal. We look forward to your positive response.